# PeerJ

# Speeding up all-against-all protein comparisons while maintaining sensitivity by considering subsequence-level homology

Lucas D. Wittwer[1,2,3,4], Ivana Piližota[1,4], Adrian M. Altenhoff[1,2,3] and Christophe Dessimoz[1,2]

[1] University College London, London, United Kingdom
[2] Swiss Institute of Bioinformatics, Zurich, Switzerland
[3] ETH Zurich, Department of Computer Science, Zurich, Switzerland
[4] These authors contributed equally to this work.

## ABSTRACT

Orthology inference and other sequence analyses across multiple genomes typically start by performing exhaustive pairwise sequence comparisons, a process referred to as "all-against-all". As this process scales quadratically in terms of the number of sequences analysed, this step can become a bottleneck, thus limiting the number of genomes that can be simultaneously analysed. Here, we explored ways of speeding-up the all-against-all step while maintaining its sensitivity. By exploiting the transitivity of homology and, crucially, ensuring that homology is defined in terms of consistent protein subsequences, our proof-of-concept resulted in a $4\times$ speedup while recovering $>99.6\%$ of all homologs identified by the full all-against-all procedure on empirical sequences sets. In comparison, state-of-the-art $k$-mer approaches are orders of magnitude faster but only recover 3–14% of all homologous pairs. We also outline ideas to further improve the speed and recall of the new approach. An open source implementation is provided as part of the OMA standalone software at http://omabrowser.org/standalone.

Corresponding author
Christophe Dessimoz,
c.dessimoz@ucl.ac.uk

# INTRODUCTION

Advances in genome sequencing have led to an immense increase in the number of available genomes (*Metzker, 2009*; *Pagani et al., 2012*). As the experimental annotation of these sequences would be prohibitively slow and expensive, there is a strong interest in computational methods (reviewed in *Rentzsch & Orengo, 2009*). Most approaches start by identifying genes of common ancestry within and across species—sets of "homologous" genes (*Patterson, 1988*). As further refinement, homologous proteins, which can be split up into paralogs and orthologs, diverged from a common ancestral protein (*Fitch, 1970*). Paralogous sequences, which start diverging through gene duplication, are believed to drive function innovation and specialisation, whereas orthologous sequences, which diverged through speciation, tend to have more similar biological function (*Tatusov,*

*1997*; *Lynch, 2000*; *Altenhoff & Dessimoz, 2012*; *Gabaldón & Koonin, 2013*). Therefore, protein function can be assigned by identifying regions of orthology between a newly sequenced and an already annotated genome (e.g., *Gaudet et al., 2011*; *Altenhoff et al., 2012*). Orthology inference is also of high interest to infer phylogenetic species trees and to study the evolution of gene families (*Altenhoff & Dessimoz, 2012*).

Due to the high biological interest for orthology detection, many orthology inference algorithms and associated databases have been developed (reviewed in *Dessimoz et al., 2012*; *Sonnhammer et al., 2014*). Some of these can be large, with databases comparing thousands of genomes of different species: eggNOG with currently 3686 genomes (*Powell et al., 2013*), Roundup with 2044 (*DeLuca et al., 2012*), OMA with 1613 genomes (*Altenhoff et al., 2011*) and OrthoDB with 1425 genomes (*Waterhouse et al., 2013*).

To detect homology, these resources rely on all-against-all protein comparison. All pairs of sequences are aligned using BLAST (*Altschul et al., 1997*) or dynamic programming (*Smith & Waterman, 1981*). They are then inferred as homologous if their alignment score is above a certain threshold and alignment length constraints are satisfied (*Dessimoz et al., 2005*; *Roth, Gonnet & Dessimoz, 2008*). Even orthology databases based on reconciled gene/species trees (*Vilella et al., 2008*) or hidden Markov model profiles (*Mi, Muruganujan & Thomas, 2013*) typically start their procedure with an all-against-all step, before proceeding to clustering.

However, the all-against-all procedure scales quadratically in the number of sequences compared and hence rapidly becomes very costly. For instance, the all-against-all phase in OMA, the resource developed in our lab, has already consumed over 6.1 million CPU hours to date, despite the use of a highly optimised implementation (*Szalkowski et al., 2008*) of the *Smith & Waterman (1981)* pairwise alignment algorithm. Furthermore, in the all-against-all phase of OMA, about half of the time is spent on unrelated gene pairs (see Supplemental Information 1) which are not used for further orthology inference.

Previous methods have been proposed to speedup the all-against-all phase, but they tend to come at a cost of lower sensitivity. CD-HIT (*Li & Godzik, 2006*), UCLUST (*Edgar, 2010*), and kClust (*Hauser, Mayer & Söding, 2013*) are methods to reduce the complexity by clustering the dataset based on the number of common or high-scoring words ($k$-mers) in each sequence. This works well to identify closely related sequences—e.g., close paralogs or "redundant" sequences—but such approaches are less sensitive than full dynamic programming alignment and thus tend to miss distant homologs. Another approach was proposed in the Hieranoid method (*Schreiber & Sonnhammer, 2013*): by using the species tree as guide and iteratively merging terminal sister taxa into ancestral genomes, the computational cost only grows linearly with the number of genomes. However, reconstructing ancestral sequences or profiles can prove challenging among distantly related species. Moreover, the guide tree must be clearly defined and known, which is a problem for many prokaryotes and eukaryotes.

Here, we present a new strategy for speeding up the all-against-all step, by first clustering sequences and then performing an all-against-all within each cluster. Crucially, in the clustering phase, the new approach accounts for residue-level homology and not

merely whole protein-level homology. As a result, sequences can belong to multiple clusters. On sets of empirical bacterial and fungal protein sequences, our new approach achieves a reduction in computational time of ~75% while maintaining the sensitivity of all-against-all procedure (>99.6% of all pairs identified by all-against-all).

## MATERIALS AND METHODS

We start by motivating our method under the assumption of perfect input data, then present the modifications required to perform well on real, imperfect data. We then describe the $k$-mer methods we used as points of comparison. Finally, we present the empirical datasets used to evaluate the new method.

### Building homologous clusters from perfect data

Typical downstream analysis of the all-against-all step requires an alignment score for every significant pair of proteins (putative homologs). Therefore, aligning these significant pairs seems difficult to avoid. In contrast, avoiding the alignment of unrelated pairs can potentially reduce the overall runtime of the all-against-all. One way to avoid some pairs is by using the transitive property of homology: typically—we discuss pathological cases in the next section—if sequence A is homologous to sequence B, and sequence B is homologous to sequence C, then sequences A and C are homologous. A corollary of this is that if sequence A is homologous to sequence B, and sequence A is *not* homologous to sequence C, then sequences B and C are *not* homologous.

*Proof of corollary*: Let S1 denote "A is homologous to B", S2 "B is homologous to C" and S3 "A is homologous to C". The transitivity of homology states that S1 $\bigwedge$ S2 $\Rightarrow$ S3. By material implication, that it is equivalent to $\neg$(S1 $\bigwedge$ S2) $\vee$ S3. De Morgan's law yields the equivalent expression ($\neg$S1 $\vee$ $\neg$S2) $\vee$ S3 which is, due to the associativity and commutativity of disjunction, equivalent to ($\neg$S1 $\vee$ S3) $\vee$ $\neg$S2. Using the De Morgan's law again, the expression transforms to the equivalent $\neg$(S1 $\bigwedge$ $\neg$S3) $\vee$ $\neg$S2. Finally, by material implication it is equivalent to the expression S1 $\bigwedge$ $\neg$S3 $\Rightarrow$ $\neg$S2.

Therefore, in this particular scenario, if we infer that A and B are homologous and A and C are non-homologous, there is no need to align sequences B and C. Another way to consider this situation is that A acts as a representative for its homologous sequence B. We can generalise this idea to more than three sequences by building clusters of homologs for which a single sequence acts as a representative for all members (trivial proof by induction).

Accordingly, we can split the all-against-all into two steps: (i) building homologous clusters, and (ii) performing the all-against-all within each cluster. In the clustering step, we initialise the procedure with a single cluster containing the first sequence as its representative and sequentially consider all the other sequences: each sequence is aligned to the representative sequence of each homologous cluster. If an alignment score is significant, implying that the sequence is homologous to members of that particular cluster, the sequence is added to the cluster. Else, a new cluster is formed with that sequence as representative.

**Peer**J

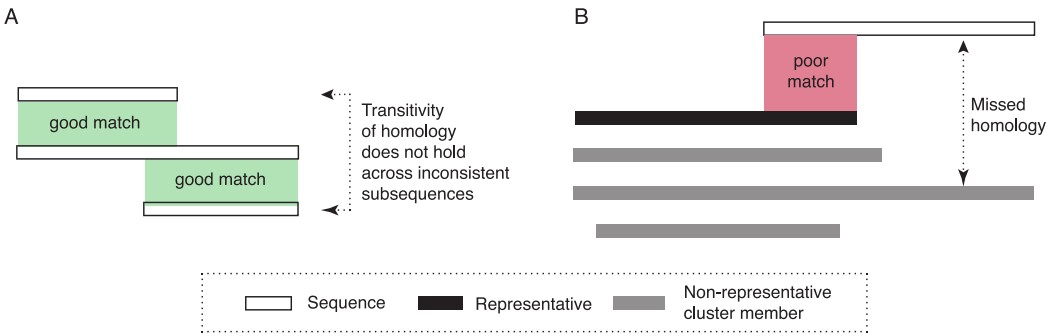

**Figure 1** **Diagram of potential problems with transitivity of homology.** (A) The transitivity of homology does not hold if the residues involved are inconsistent. (B) Homologous relationships can be missed if significant sequence parts are not covered by the representative sequence. To address this problem, our approach would create a new cluster, with the sequence that is not fully covered as its representative.

Under the assumptions that homology is transitive and that homology can be perfectly ascertained through sequence alignment, this procedure guarantees that all homologous sequences end up in a common cluster in $O(nk)$, where $n$ is the number of proteins and $k$ is the number of clusters. Since $k \leq n$ (and possibly $k \ll n$), this is more efficient than the $O(n^2)$ runtime of the standard all-against-all procedure.

In practice, however, homology at the level of proteins is not always transitive and homology cannot be perfectly ascertained. Therefore, refinements to the basic idea are required.

## Building homologous clusters from real data: multiple representatives, multiple clusters, and subsequence-level homology

The two assumptions underlying the algorithm for ideal data are not tenable on real data. First, highly distant (and thus dissimilar) homologous sequences can be very difficult or impossible to reliably identify. Thus, if the representative is very distant to a particular homologous sequence, homology can nevertheless be missed. Second, homology does not necessarily apply uniformly across all residues of a protein. For transitivity to hold, it is essential that homology applies to a common set of residues. However, insertions and gene fusion can lead to situations where this is not the case (see Fig. 1A for an illustrative example).

Thus, to deal more effectively with real sequences, we modify the basic algorithm in three ways. First, instead of a single representative sequence, we also consider a variation of the algorithm that uses several representative sequences for each cluster. This can be advantageous for homologous clusters containing very dissimilar sequences, which are thus poorly represented by any single sequence. Second, we allow for the possibility that some sequences might be homologous to more than one cluster. This could be the case if homologous clusters are fragmented due to high divergence among some of their members (and among their representatives in particular). This also enables multi-domain proteins to be included in clusters corresponding to different domains. Third, we take into account

subsequence-level homology: we ensure that the entire length of a particular protein (minus a tolerance parameter, set to 20 amino-acid residues in this study) is covered by the cluster(s) to which the sequence belongs. If this is not the case, a new cluster is created with the sequence as representative. Note that this is in addition to inclusion into any cluster with significant alignment (Fig. 1B).

Putting these ideas together, our approach for building homologous clusters is as follows. First, we sort the input genomes by their number of sequences in descending order. For every genome, starting with the one with the highest number of sequences, we process its sequences one by one as they are listed in the database file. As there are no clusters at the beginning of the process, the first sequence founds a cluster and becomes its representative. Every subsequent sequence is aligned with the representatives of the current cluster(s) using *Smith & Waterman (1981)*. We run the vectorised Smith–Waterman implementation of *Szalkowski et al. (2008)* using the 224 GCB scoring matrix (*Gonnet, Cohen & Benner, 1992*). If the score is above the minimum threshold of 135.75, the sequence is added to the cluster. Furthermore, if the number of cluster representatives in the particular cluster is below the maximum allowed, the newly added sequence becomes a representative. We also keep track how much the sequence is covered by the representatives of the assigned clusters. We do not introduce any restrictions on the size of the clusters nor on the number of clusters that a sequence is assigned to. After an exhaustive search through all cluster representatives, we assess whether the sequence was added to one or multiple clusters. If the full length of the sequence (minus a tolerance of 20 amino acids) is not covered by the clusters to which the sequence was added, an additional cluster is created with the sequence as a representative. The same applies if the sequence was not assigned to any clusters (see Fig. 2 for pseudocode).

## Computing all-against-all within each cluster

Once we have identified all homologous clusters, we compute an all-against-all within each cluster using the same criteria as the global all-against-all (Fig. 3). This ensures that all retained pairs agree with the score and length requirements of the full all-against-all, and provides scores for all homologous pairs, which are often required for downstream analyses.

## Comparison with full all-against-all and other methods

The baseline of this study is given by the current all-against-all procedure in the OMA database (*Roth, Gonnet & Dessimoz, 2008*; *Altenhoff et al., 2011*). It runs a highly optimised *Smith & Waterman (1981)* local alignment implementation (*Szalkowski et al., 2008*) using the 224 GCB scoring matrix (*Gonnet, Cohen & Benner, 1992*). To be inferred as homologs, sequence pairs need to reach an alignment score of at least 135.75 with the 224 GCB matrix (and 181 with an optimised GCB matrix, but this additional requirement was not considered in this study), with the additional requirement that the length of the shorter sequence aligned be at least 61% of the longer sequence. These parameters were previously established to yield good results on empirical data (*Roth, Gonnet & Dessimoz, 2008*).

```
Input:  Genomes , thresholds T=135.75 (minimum score), L (number of representatives per cluster) and
        C=20 (minimum coverage)

Clusters ← ∅
for each genome in Genomes do
        for each sequence in genome do
                if Clusters = ∅ then
                        initialise newCluster
                        newCluster ← sequence
                        newCluster.representatives ← sequence
                        Clusters.append(newCluster)
                else
                        for each cluster in Clusters do
                                for each representative in cluster.representatives do
                                        if alignment-score(sequence, representative) > T then
                                                cluster.append(sequence)
                                                if |cluster.representatives| < L then
                                                        cluster.representatives ← sequence
                                                end if
                                        end if
                                end for
                        end for
                end if
                if sequence was assigned to at least one cluster then
                        coverages ← ∅
                        for each representative of the assigned clusters do
                                coverages.append(coverage(sequence, representative))
                        end for
                end if
                if sequence was not assigned to any cluster or sumlengths(coverages.uncovered_parts()) > C then
                        initialise newCluster
                        newCluster ← sequence
                        newCluster.representatives ← sequence
                        Clusters.append(newCluster)
                end if
        end for
end for

Output: a set of homologous clusters Clusters
```

**Figure 2  Pseudocode of the proposed cluster building procedure.**

To compare the performance of the new approach to $k$-mer clustering, we also ran kClust (*Hauser, Mayer & Söding, 2013*) and UCLUST (*Edgar, 2010*). We ran kClust with default set of parameters, where the required minimum sequence identity was 30% (`-s 1.12`) and the minimum alignment coverage of the longer sequence was 80% (`-c 0.8`). To increase sensitivity, we also ran kClust with 30% as minimum for sequence identity (`-s 1.12`) and 50% as minimum alignment coverage (`-c 0.5`). The jobs were run on a single core of a Dell R910 (Intel Xeon E7-8837, 2.66 GHz) with 32 cores and 1TB of memory. For UCLUST, we used the recommended option `-cluster_fast`. We first ran UCLUST with default parameters, specifying 30% as sequence identity (`-id 0.3`). We also ran it with the additional requirements on coverage of the target sequence being 50% (`-target_cov 0.5`) and allowing sequences to be assigned to multiple clusters (`-maxaccepts 0`), without limiting the number of sequences rejected before quitting search (`-maxrejects 0`). In both cases, only sequences longer than 50 amino acid residues were taken into account—the same length requirement as in the all-against-all procedure. UCLUST was run on a MacBook Air (Intel Core i7, 1.7 GHz) dual-core with 8 GB of memory.

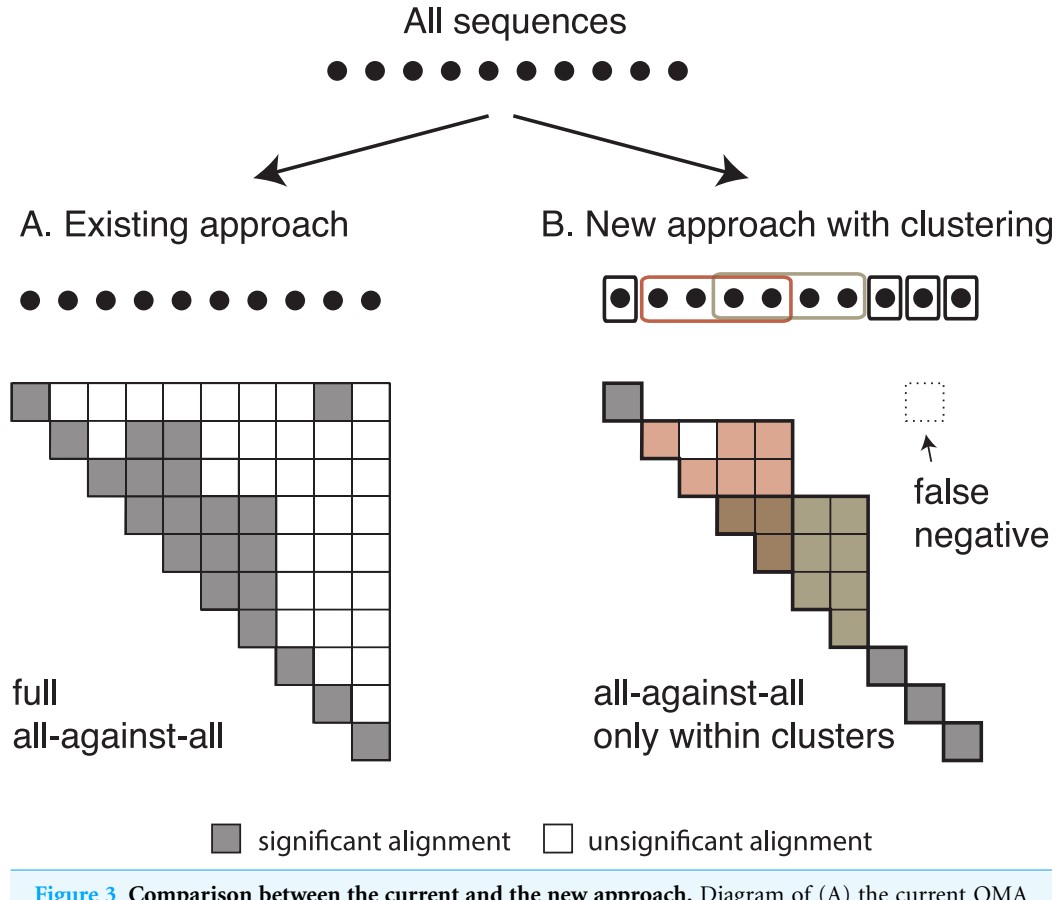

**Figure 3 Comparison between the current and the new approach.** Diagram of (A) the current OMA all-against-all step and (B) the new approach with homologous clustering and sequence coverage. In the proposed approach, the sequences are first clustered and then the all-against-all is run only within clusters. The overall number of computations is considerably reduced, but some homologous relationships can be missed.

## Datasets

We analysed the performance of the new approach on two distinct datasets: bacteria and fungi. Bacteria dataset contained 14 proteomes, fungi contained 12, all sampled from the OMA database (*Altenhoff et al., 2011*) (March 2014 release; see File S1 for complete list). The distribution of sequence lengths and evolutionary distances among significant pairs is provided in Figs. S1–S2. To assess the scaling behaviour of each variant, we ran them on subsets which were formed as follows: all genomes in the dataset were sorted by their size and the ones in the middle of the list were chosen to form a subset. By choosing the central 2, 4, 6, 8, 12 and finally all 14 proteomes, subsets of bacteria were constructed. We repeated the procedure for fungi and obtained subsets of size 2, 4, 6, 8, 10 and 12 proteomes.

In addition, we performed an extra analysis on a diverse dataset containing two vertebrates, one plant, one protist, one fungus, and one bacterium (File S1).

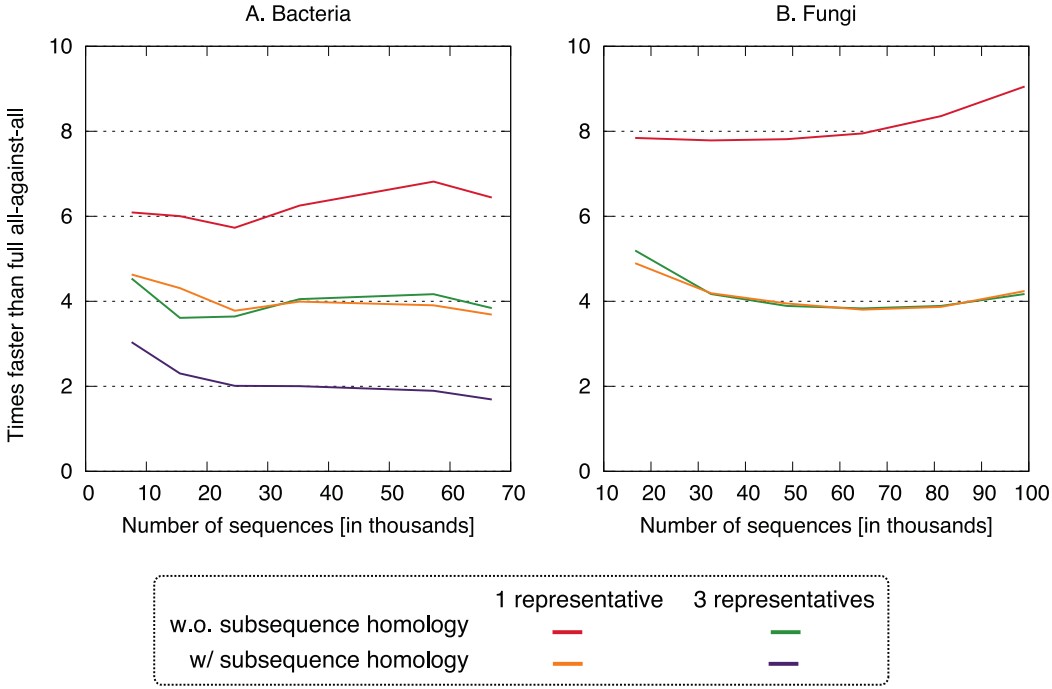

**Figure 4 Speed up using the new approach.** The proposed approach is 2–9× faster, depending on datasets and variants. On the fungi dataset, only the three fastest variants were computed.

## RESULTS

To assess the time and accuracy of our new approach, we compared the four clustering variants—one or three cluster representatives, each combined with either taking into account residue-level homology or not (see Methods)—on empirical bacterial genomes. We also repeated the three fastest variants on empirical fungal datasets.

All four clustering variants decreased runtime compared to the full all-against-all algorithm, with a speedup factor of 2–9 depending on the variant and dataset (Fig. 4). Unsurprisingly, the reduction was strongest in variants considering a single representative sequence per cluster. Likewise, ignoring residue-level homology lead to faster runtime, though the difference was more modest.

In terms of performance—which we measured in terms of the proportion of significant pairs from the full all-against-all that were recovered by the new variants (referred to as "recall") or its complement (referred to as "missing pairs")—we observed a clear benefit in taking into account residue-level homology, with recall values consistently >99.6% (Fig. 5). The use of three representative sequences per cluster also led to increase in recall, but not nearly to the same extent. The recall values obtained for all variants were consistent across the bacteria and fungi datasets and for different sizes, which suggests that they might hold across other datasets as well. To illustrate the nature of missing homologs, two cases are detailed in Figs. S3–S6.

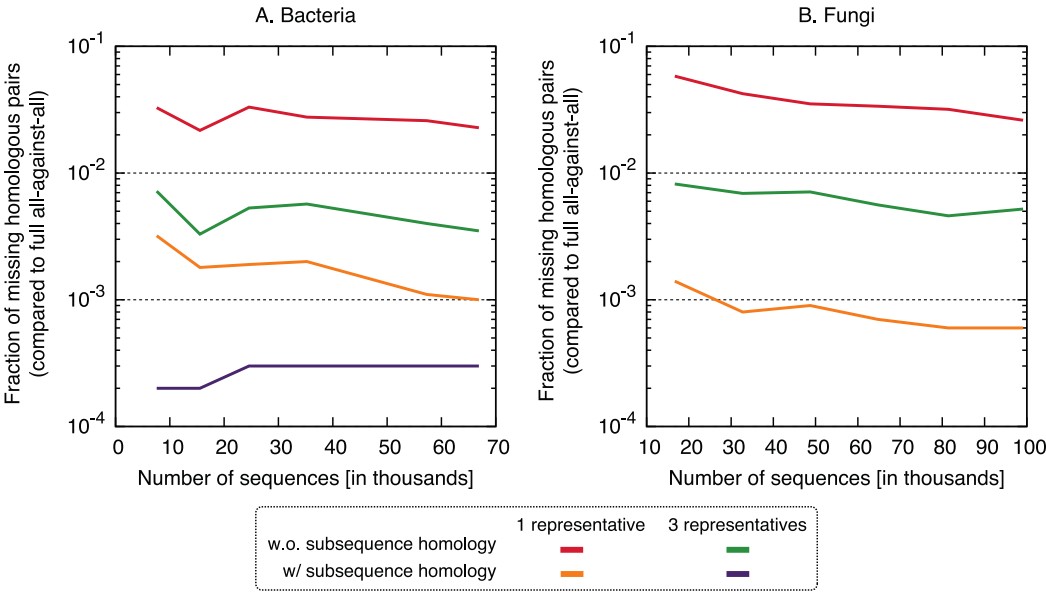

**Figure 5 Fraction of the pairs missed when using the proposed algorithm.** The new approach misses 0.4–6% of the pairs from the full all-against-all in its simple variant (1 and 3 representatives), and 0.01–0.3% taking into account sub-sequence homology.

Putting runtime and accuracy together, the best results were obtained by taking into account subsequence-level homology using a single representative sequence, which achieved a ~4× speedup while maintaining >99.6% recall in both datasets.

To gain insight into the nature of missing pairs, we analysed the distribution of alignment scores of missing pairs compared to that of all significant matches (Fig. 6). For all method variants, but especially those taking into account subsequence-level homology, the distribution is heavily skewed toward low-scoring matches. This is favourable, because low-scoring matches are less reliable in the first place and therefore downstream analyses are already designed to cope with missing homologous relationships among distant pairs (e.g., *Altenhoff et al., 2013*). We also note that the higher recall observed with three representatives instead of one is mainly concentrated on low scoring matches. Furthermore, a per-gene analysis of the rate of missing pairs suggests that errors are distributed quite evenly across all family sizes (Fig. S7).

For any given taxonomic range, adding new genomes and the corresponding new sequences can only increase the number of clusters. However, as the number of genomes grows, we can expect that an increasing proportion of the sequences will fall into one of the existing clusters. Thus, we should see a tapering in the number of clusters as a function of the number of sequences, which would be favourable in terms of runtime of the algorithm. In our datasets, we could not observe such tapering, and instead the growth in cluster numbers was broadly linear (Figs. S8–S9). This suggests that 12–14 genomes are too few to discern the additional asymptotic benefits of our new approach. A similar conclusion can be drawn from the distribution of cluster size, which is heavily skewed toward very small clusters (Fig. S10). The large overlap among numerous clusters and the existence of

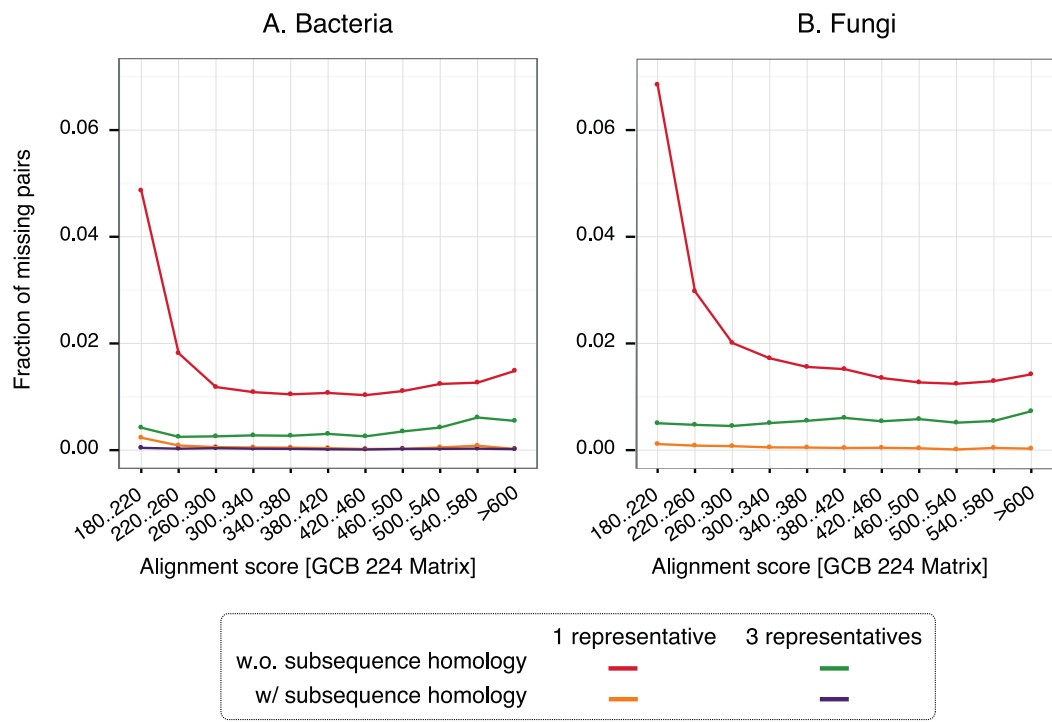

**Figure 6 Distribution of alignments scores of missing pairs.** With a single representative, the distribution is heavily skewed toward low-scoring pairs.

many sequences included in multiple clusters (Fig. S11) suggests improvement potential by merging some of the clusters (see also *Discussion* below).

To assess the impact of the new clustering approach on orthology inference, we ran the OMA standalone software (*Altenhoff et al., 2013*) using our best new variant (1 representative, with subsequence homology) compared to the full all-against-all as a reference. On the largest Bacteria and Fungi dataset, the proportion of orthologs that were recovered was 99.71% and 99.87% respectively. This is slightly lower than the proportion of recovered homologs (99.9% and 99.94%), but remains very high.

Furthermore, to test whether large genomes and/or multidomain proteins could affect the performance of the new approach, we conducted an additional comparison on a genome set containing two vertebrates (Human and *Xenopus tropicalis*), one plant (*Arabidopsis thaliana*) and three unicellular organisms (*Plasmodium falciparum, Saccharomyces cerevisiae*, and *Bacillus subtilis*). We applied the variant with one representative and subsequence homology. It resulted in a considerably higher speedup factor of 12.05, with a recall of 99.94%. This suggests that the new approach behaves well in the presence of large genomes and numerous multi-domain proteins as well.

Finally, we sought to compare the new approach to the state-of-the-art *k*-mer clustering approaches UCLUST (*Edgar, 2010*) and kClust (*Hauser, Mayer & Söding, 2013*). As elaborated in the introduction, these methods have been developed to cluster closely related sequences, but given their similar conceptual ideas (clustering, use of

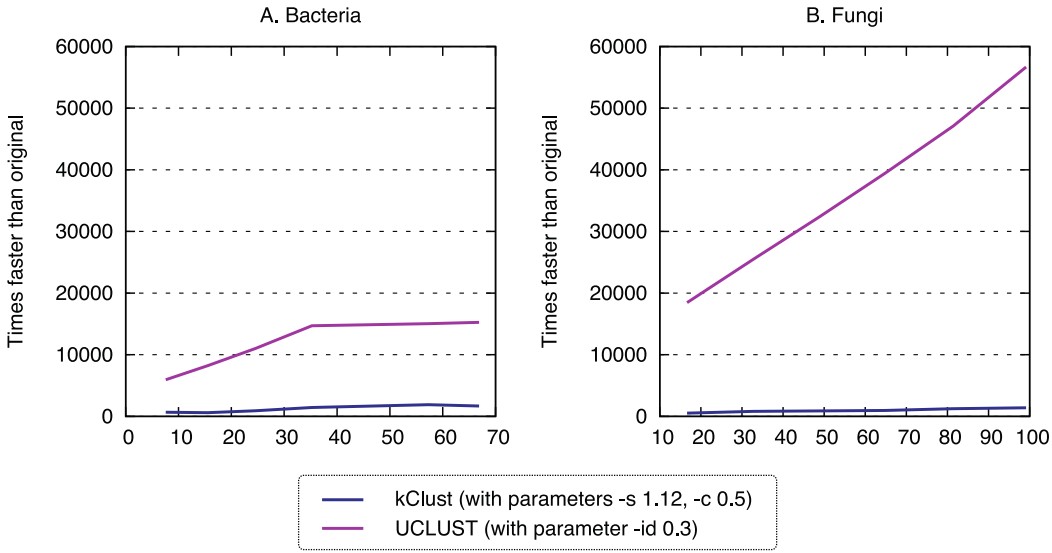

**Figure 7 Runtime of typical K-*mer* approaches.** kClust and UCLUST are several orders of magnitude faster than full all-against-all.

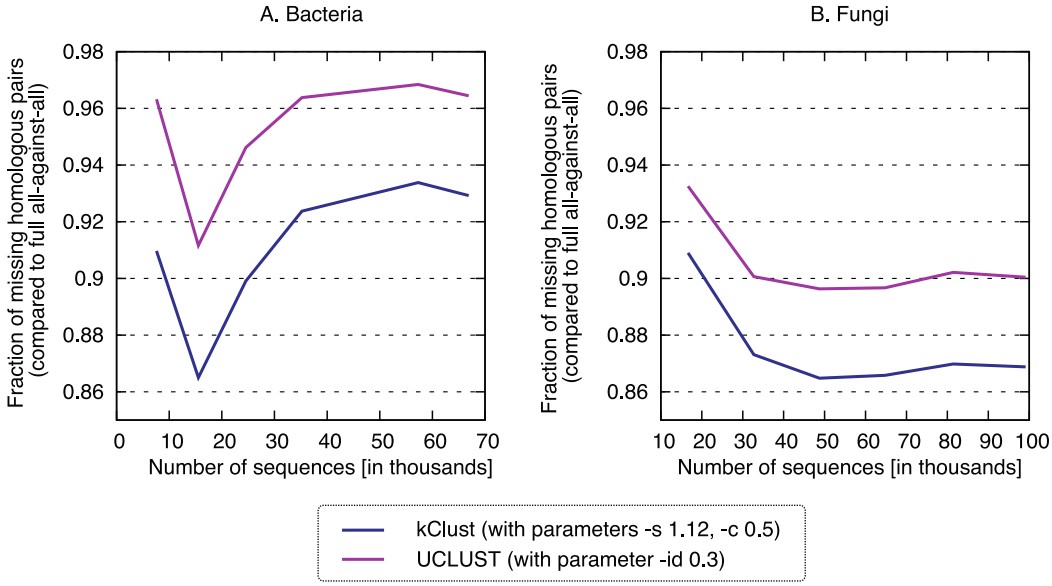

**Figure 8 Recall of typical K-*mer* approaches.** kClust and UCLUST only recover 3–14% of the homologous pairs identified by the full all-against-all procedure, even when they are run with sensitive parameter settings (see Materials and Methods for details).

representative sequences), they provide an interesting point of comparison. As expected, these approaches were orders of magnitude faster (speedup 500–1900× for kClust, over 6000× for UCLUST; Fig. 7), but their recall was typically in the single percentage points (6.62–13.52% for kClust, 3.16–10.37% for UCLUST), even with inclusive parameters (Fig. 8; File S2).

## DISCUSSION AND OUTLOOK

The all-against-all phase is at the basis of many orthology inference algorithms but due to its quadratic time complexity in terms of number of sequences, the all-against-all phase can be a bottleneck. This work suggests that the transitivity of homology can be exploited to substantially speedup homology inference while maintaining sensitivity—provided homology is defined in terms of consistent subsequence stretches.

The datasets used in this study are challenging, with median distance of 146 PAM (1.46 substitution per site on average) in the bacteria dataset and 149 PAM in the fungi dataset (Fig. S2). This implies that half of homologous pairs have less than 30% sequence identity. At such high levels of divergence, *k*-mer based methods perform poorly compared to all-against-all dynamic programming alignment. In contrast, our transitivity clustering approach performed well. It is likely to perform even better on evolutionarily closer sets of taxa, such as vertebrate species or flowering plant species.

The speedup we observed with the new approach is substantial. But because the number of clusters increased roughly linearly with the number of genomes in our datasets of up to 14 genomes, the overall time complexity still grows quadratically in this range. However, as rarefaction curves show (e.g., *Mira et al., 2010*), the growth in the number of homologous clusters typically tapers off. Thus, it is possible that the asymptotic complexity of the new procedure is subquadratic in the number of species. This will need to be confirmed in future work.

Apart from the full dynamic programming alignment comparison with cluster representatives and the subsequence-level homology consideration, our approach distinguishes itself from most clustering methods in that it allows sequences to match to several clusters. If homology was always uniform across the whole sequence length and could be perfectly inferred from present-day sequences, each sequence would belong to one and only one homologous cluster. In practice however, these assumptions do not hold: gene fusion can result in sequences that have homologous relationships across multiple clusters; some homologs have diverged too far to be inferred as such, thus resulting in cluster fragmentation. By allowing for sequences to belong to multiple clusters, our method is robust to these complications.

Though the present study focuses on Smith–Waterman dynamic programming alignments, a similar clustering approach would be possible with the faster but less sensitive BLAST (*Altschul et al., 1997*). One complication with BLAST is however that the sequence database needs to be re-indexed whenever a sequence is added. To mitigate this, one could add new representative sequences in batches, with the additional complication that sequences within each batch would also need to be aligned to one another.

A further advantage of the clustering approach is that it also works well in the context of "semi-curated" databases, such as in COGs (*Tatusov et al., 2003*), Panther (*Mi, Muruganujan & Thomas, 2013*), or PFam (*Finn et al., 2014*). Indeed, it is conceptually straightforward to let curators optimise particular homologous clusters by fine-tuning representative sequences, coverage and score thresholds on a cluster-by-cluster basis.

These promising results notwithstanding, the new approach still has much potential for improvement. To further improve the recall, several ideas could be explored. First, the procedure could merge clusters that have a high proportion of members in common. This could recover some of the missed homologous pairs. Second, the choice of cluster representatives could be improved. Indeed, the current strategy of selecting the first sequence (or the first three sequences) added to the cluster is likely to be suboptimal in most instances. Instead, a better choice of representative would be to try to select a sequence with minimal average distance to all cluster members (a "centroid" sequence). Another idea, suggested by Dr. Ikuo Uchiyama in his peer-review, would be to vary the number of representatives depending on the particular needs of each cluster, e.g. by adding a new member as a representative if its similarity score with the existing representative(s) is below a certain threshold. Third, as an alternative to representative sequences, clusters could be represented by Hidden Markov model profiles. Such approach, at the basis of well-established methods such as PSI-Blast (*Altschul et al., 1997*), HMMER (*Eddy, 2011*), or HHBlitz (*Remmert et al., 2011*), is likely be more sensitive than pairwise alignment with a single representative sequence. The profiles could be periodically updated as new sequences are assigned to the cluster.

We also see potential to further improve the speed of the new approach. First, merging homologous clusters can lead to a speed improvement because the cost of assigning sequences to clusters grows linearly in the number of clusters. This needs to be done carefully, because excessive merging—the merging of clusters containing a substantial number of non-homologous pairs—can reduce the efficiency of the within-cluster all-against-all, whose time complexity grows quadratically in the number of sequences. Second, it may be possible to optimise the assignment of sequences to clusters by identifying clusters that are so different to one another that they are practically mutually exclusive and thus inclusion into one implies exclusion from the other. An empirical way of establishing such mutual exclusivity would be to keep track of the number or proportion of sequences belonging to both clusters. Finally, the new approach could be parallelised. One way of parallelising the assignment of sequences to clusters would be to use a Publisher-Subscriber model (*Eugster et al., 2003*): a "master" process would start the analysis of a new sequence by distributing it to a set of "workers", each responsible to compare the sequence to a subset of all existing clusters. Each worker would thus align the new sequence to its designated subset of clusters and reports back significant matches and their associate sequence ranges (subsequence-level coverage). Once the master process has received this information from all workers, it would ensure that the new sequence is fully covered by matches to existing clusters, and else it would generate a new cluster with that sequence as representative. As for the within-cluster all-against-all comparisons, they could be straightforwardly parallelised thanks to the lack of dependency among all pairs.

Meanwhile, a serial implementation of our best variant (accounting for subsequence-level homology using a single representative sequence) is available as part of the open source OMA standalone package (http://omabrowser.org/standalone).

## ACKNOWLEDGEMENTS

This article is dedicated to Gaston H. Gonnet on the occasion of his retirement. We thank Kevin Gori, Henning Redestig, Nives Škunca, Bartlomiej Tomiczek for useful feedback on the manuscript, as well as Odile Lecompte, Benjamin Linard, Ikuo Uchiyama, and one anonymous reviewer for useful feedback on the manuscript.

### Funding

AMA is funded by a Swiss Institute of Bioinformatics Infrastructure Grant. IP is jointly funded by a UCL Impact Award and by Bayer CropScience. Open access publication charges are covered by the University College London Library. The funders had no role in study design, data collection and analysis, decision to publish, or preparation of the manuscript.

### Grant Disclosures

The following grant information was disclosed by the authors:
Swiss Institute of Bioinformatics "Service and Infrastructure Grant".
UCL Impact Award.
Bayer CropScience.
University College London Library.

### Competing Interests

CD is an Academic Editor for PeerJ and co-organises the GNOME 2014 symposium for which this submission is a contribution.

### Author Contributions

- Lucas D. Wittwer and Ivana Piližota performed the experiments, analyzed the data, contributed reagents/materials/analysis tools, wrote the paper, prepared figures and/or tables, reviewed drafts of the paper.
- Adrian M. Altenhoff analyzed the data, contributed reagents/materials/analysis tools, wrote the paper, prepared figures and/or tables, reviewed drafts of the paper.
- Christophe Dessimoz conceived and designed the experiments, analyzed the data, contributed reagents/materials/analysis tools, wrote the paper, prepared figures and/or tables, reviewed drafts of the paper.

### Supplemental Information

Supplemental information for this article can be found online at http://dx.doi.org/10.7717/peerj.607#supplemental-information.

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
