# Peer review of "Speeding up all-against-all protein comparisons while maintaining sensitivity by considering subsequence-level homology"

_PeerJ, doi:10.7717/peerj.607_

## Round 0.1 · original submission · Major Revisions

· Academic Editor

Major Revisions

The reviewers all agree that this is a potentially important paper for the field. However, there are some serious concerns about the level of detail at which the methods are presented; reviewers 1, 2 and 4 discuss this the best. Reviewer 3 has also raised a concern about the diversity of the datasets; this needs to be discussed at least.

·

Basic reporting

The method presented in this paper is of great interest for the field of comparative genomics and even bioinformatics in general. Today, the generation of a blast all-against-all, compulsory step for all large-scale orthology inference algorithms, is a real challenge because new genomes are daily released. The introduction describes this challenge in a clear way.
Consequently, I was very excited by the methodological innovations proposed by the authors and the important computational improvements that their results are demonstrating. Generally, I think this is a good manuscript but before publication, I think that some major points should be more extensively detailed by the authors. I explain my concerns below.

Experimental design

My major issue is the lack of details in the methods. I think that currently I would not be able to fully understand/reproduce the main steps of the new methodology, in particular for the following points:

1 Initial cluster building.

How are chosen the initial representative sequences? Is it a random choice? Are they sequence represented by a minimum of dissimilarity? The same number of sequences per organism ? no details are provided by the authors.

2. Cluster extension

How is exactly decided the cluster extension, with which metrics?
When and how the end of the extension of a cluster is decided?

Such information should be introduced in the methods, making the cluster formation approach a bit less obscure. Compared to the classical blast all-against-all, the main development of the new method seems to be this preliminary clustering step. Consequently, I was disappointed to not really understand this step in details. Similarly, when the new methodology is compared to kClust and UCLUST , the authors describe in details the sequence similarity parameters used with this programs, but not for their own method.

Validity of the findings

It would be interesting to describe the content of the preliminary clusters. Does some clusters aggregate only very similar sequences while some other contain more divergent proteins?

Concerning the false negative homologous pairs, some examples could be discussed to better highlight their origin. Does some of these low score matches are due to the nature of the sequence itself? (short repetitive domains, low complexity regions) or the fact that they are at long evolutionary distance (basically, is there more false positive between distant bacteria/fungi than between than between large divergent protein families in closely related organisms). This is a complicated subject and new results don’t necessarily need to be produced for this point. But debating the nature of the false negative in the Discussion would be an interesting addition.

To complete the bacterial and fungal datasets, a 3rd dataset showing the behaviour of the algorithm with hundreds of very distant organisms could be another addition.
The chosen datasets, fungal and bacterial proteomes, are nice examples but no homology is generally found between huge parts of their proteomes. 30% of their ORFs (and consequently proteins) are generally highly specific elements, not shared between genera (between saccharomyces strains already hundreds of ORFs are specific, see Stacia R. Engel* and J. Michael Cherry, 2014 ; up to 30% between candida and saccharomyces, Edward L. Braun et al., 2000 Genome Res. ). Consequently, if I understand it well, the sentence “This implies that half of homologous pairs have less than 30% sequence identity.” is obvious. What about studying an organism set which includes both such kind of proteome and the more conserved proteomes of animals (where few organism-specific sequences are observed)? For instance, a mix of 100 fungi/plants/protest/animal proteomes (with balanced phyletic composition) ? Indeed, the real problem today is the generation of a blast all-against-all for hundreds of proteomes, not 12.
The paper would be more strong if similar speed improvements are shown for such a dataset. I understand that generating such dataset is a long process and could require weeks of calculations. Consequently, I ask the editor to consider this last point as a strong suggestion but not a required revision.

Comments for the author

I think that the results demonstrated in the manuscript are very promising. But, as the paper describes a new method based on a preliminary sequence clustering, then followed by a all-against-all comparison in each cluster, I really recommend a better description of this initial step, which seems to be the key to the large speed improvement.

·

Basic reporting

The manuscript is well written. The problem to be addressed is important in this field and the basic idea is interesting.

I have only one minor comment:

1) In figure legend, each figure should be titled (with one sentence description).

Experimental design

I have two comments regarding the exact range of the research question considered here:

1) The purpose of this work is somewhat unclear. I think one of the fundamental problems in all-against-all comparison is its quadratic time scale. However, although the authors repeatedly mentioned this point, and their method have some potential to address it, I think the problem of quadratic time scale is not clearly solved in this paper. Therefore if this is a fundamental question of this paper, I must say that the experimental design is not adequate.

2) It seems that their method is suitable for rigorous Smith-Waterman algorithm but may not fit well to the BLAST like method where database indexing is needed before search because in their method the target database is repeatedly changed, which requests database re-indexing and reduces the efficiency. Given that BLAST is one of the most commonly used tools for this kind of analysis, the authors should mention to the applicability of their method to BLAST, and if not applicable, should compare their method based on Smith-Waterman algorithm with a simple all-against-all BLAST search.

One additional minor comment:
3) The definition of "Reduction in time" in Fig. 3 and 6 is somewhat unclear to me. I think that "80% reduction" is equivalent to "relative computational time is 0.2" but the latter is more intuitive to me since it is easily converted into more intuitive statement "5 times faster than the original".

Validity of the findings

I have two significant comments.

1) Difference of the fractions of missing homologs among methods (Fig4 and 7) is clearly demonstrated, but I think its effect is not clear. Since all-against-all search is used for some sort of clustering method (such as OMA), the method should also be evaluated whether the clustering result based on the modified all-against-all comparison is close enough to that based on the original all-against-all comparison. The result should be different among the clustering method that is intended to be applied (depending on the granularity of clustering and treatment of multi-domain proteins etc.), so the authors should discuss this point.

2) As a multiple-representative strategy, the authors considered only 3-representative strategy, which was excluded from their best strategy because of its inefficient runtime (page 9), but just discarding this strategy leaves the first concern mentioned at the beginning of section 2.2 (page 4) unsolved. In fact I cannot understand why they consider only a fixed number of representatives. Since it is natural that a larger cluster has more representatives, why not consider a strategy of adding a new representative sequence only when it is not similar to any existing representative with sufficiently high score.

One additional minor comment:
3) In Fig 5, I think displaying the fraction of missing pairs instead of the number of missing pairs in each score range on y-axis can clarify the result.

·

Basic reporting

No comments

Experimental design

The datasets used in this study should be extended and diversified. As mentioned by the authors, there is a linear growth of the number of clusters with the number of genomes considered. It is therefore difficult to assess the real benefit of the approach in the treatment large datasets. Moreover, the fungi dataset mainly consists of Ascomycota. The approach should be tested on a larger range of eukaryotic sequences including more divergent sequences and in particular, sequences from Metazoan and plants. Proteins from higher eukaryotes often show a mosaic domain composition and must be taken into account to evaluate the robustness of the sub-sequence homology approach.

Validity of the findings

It could be valuable to see the effect of the different parameters on the number of clusters, the fraction of proteins involved in several clusters and the fraction of “overlapping” clusters.

Comments for the author

The proposed method addresses a crucial problem in the field of orthology inference and more generally in comparative genomics. The strategy based on the transitivity of homology takes into account the complexity of protein evolution by considering protein subsequences and by allowing proteins to be included in several clusters. Results obtained on small subsets are promising but have to be confirmed on a larger set including highly modular proteins from large genomes of plants and animals.

Reviewer 4 ·

Basic reporting

The manuscript reveals that first clustering sequences on k-mers in smart way, gives a tremendous speed improvement with only very little sensitivity loss.

The algorithm for building homologous clusters from real data is presented in one paragraph. This paragraph describes the scenario's that the algorithm considers - which are very reasonable considerations. But given that this the key innovation of the paper, I think this section does not offer enough information to get sufficient understanding of what the algorithm is exactly doing. Maybe pseudo-code, or something like a decision tree / pipeline could be added.

This is a non-essential point which is already discussed by the authors, yet I nevertheless think that this discussion could be slightly altered. It concerns the discussion for “curated” or semi curated homology or orthology datasets (COG, Panther, etc.). The essential feature of these tools is that you can simply scan each new genome for orthologous groups and hence it scales linearly with the increase in the number of genomes, while at the same time solving the protein fusion/fission problems. Of course if one were to update e.g. COGs via complete all-vs-all then the methods of this manuscript could speed things up tremendously. But for such an important update speed would be less important than sensitivity, and more importantly such an update could start from decent orthologous groups and only concern complicated groups and “new” groups that were not present in the original species set. Such an informed strategy would drastically reduce the computational demands.

Experimental design

The method works very well. All the results are presented as fraction of true relations recovered or wrong relations inferred. I agree that this is the most important metric for this important step in orthology inference. However, I would also like to see information on a per gene level. i.e. if some genes have many true and easy to detect homologies (e.g. ATPases, GTPases,TPR-repeats), than this dominates the data set, while relevant genes that belong to small / diverged families are wrongly assigned. Could you also report the results on a per gene level, or maybe re-weight (downwards) the relations based on the number of “true” relations both genes participate to present as addiotional results?

Validity of the findings

The results are valid.

---

## Round 0.2 · accepted · Accept

· Academic Editor

Accept

Congrats! It is acceptable .